# The Scorching Truth: Investigating the Impact of Heatwaves on Selangor’s Elderly Hospitalisations

**DOI:** 10.3390/ijerph20105910

**Published:** 2023-05-22

**Authors:** Kun Hing Yong, Yen Nee Teo, Mohsen Azadbakht, Hai Phung, Cordia Chu

**Affiliations:** 1School of Medicine and Dentistry, Griffith University, Brisbane, QLD 4111, Australia; hai.n.phung@griffith.edu.au (H.P.); c.chu@griffith.edu.au (C.C.); 2Institute of Malaysian and International Studies, Universiti Kebangsaan Malaysia, Bangi 43600, Selangor, Malaysia; annteo1437@gmail.com; 3Department of Infrastructure Engineering, University of Melbourne, Melbourne, VIC 3010, Australia; m.azadbakht@fugro.com

**Keywords:** climate change, heatwaves, the elderly, hospitalisation, risk

## Abstract

Global climate change has contributed to the intensity, frequency, and duration of heatwave events. The association between heatwaves and elderly mortality is highly researched in developed countries. In contrast, heatwave impact on hospital admissions has been insufficiently studied worldwide due to data availability and sensitivity. In our opinion, the relationship between heatwaves and hospital admissions is worthwhile to explore as it could have a profound impact on healthcare systems. Therefore, we aimed to investigate the associations between heatwaves and hospitalisations for the elderly by age group in Selangor, Malaysia, from 2010 to 2020. We further explored the impact of heatwaves on the risks of cause-specific hospital admissions across age groups within the elderly. This study applied generalized additive models (GAMs) with the Poisson family and distributed lag models (DLMs) to estimate the effect of heatwaves on hospitalisations. According to the findings, there was no significant increase in hospitalisations for those aged 60 and older during heatwaves; however, a rise in mean apparent temperature (ATmean) by 1 °C significantly increased the risk of hospital admission by 12.9%. Heatwaves had no immediate effects on hospital admissions among elderly patients, but significant delay effects were identified for ATmean with a lag of 0–3 days. The hospital admission rates of the elderly groups started declining after a 5-day average following the heatwave event. Females were found to be relatively more vulnerable than males during heatwave periods. Consequently, these results can provide a reference to improve public health strategies to target elderly people who are at the greatest risk of hospitalisations due to heatwaves. Development of early heatwave and health warning systems for the elderly would assist with preventing and reducing health risks while also minimising the burden on the whole hospital system in Selangor, Malaysia.

## 1. Introduction

The warming climate has caused heatwaves (exceptionally hot periods) to begin earlier, last longer, and be more intense. The National Oceanic and Atmospheric Administration (NOAA) reported that 2011–2020 was warmer compared to the previous decade, 2001–2010 [1]. Global temperatures rise as a result of greenhouse gases accumulating in the atmosphere [2]. The Intergovernmental Panel on Climate Change (IPCC) warned that the average global temperature is projected to increase to 1.5 °C by 2052 without any concerted global action to reduce greenhouse gas emissions, which have caused an increase of 1 °C from pre-industrial levels [3]. A number of studies have demonstrated that heatwaves are increasing in frequency and intensity across all regions as a result of anthropogenic climate change [4]. Other studies found that heatwave trends in the United States (US) [5,6,7], Canada [8], Spain [9], China [10,11], Korea [12], Australia [13], and Southeast Asia [14] are soaring.

As evidence mounts, heatwaves pose a serious threat to human health [15,16,17,18,19]. The previous literature has established that elderly people are extremely vulnerable to heatwaves [20,21,22,23,24,25]. Epidemiological evidence suggests that elderly people are more likely to develop age-related disorders such as dementia or more severe chronic ailments, from diabetes to cardiovascular disease [26]. Many elderly people suffer from various medical conditions that make it difficult for them to care for themselves, access facilities, and escape dangerous situations. According to the National Council of Aging (NCOA), 95% of the elderly in the US have at least one chronic disease (e.g., circulatory disease, diabetes, and arthritis), and 80% have at least two conditions [27]. Researchers found that elderly people with diabetes tend to sweat less during heatwaves, and this can have detrimental effects on cardiovascular health and glucose control [28].

Several studies have revealed that elderly people suffering from circulatory diseases [29,30,31,32] and respiratory diseases [32,33,34,35] are more susceptible to heatwaves. Rapid urbanisation and an ageing population have also led to a growing elderly population living in cities. The WHO projected that 80% of elderly people will live in low-and middle- income countries by 2050 [36]. As urban expansion and heatwave trends increase, the elderly who have chronic diseases and live in urban areas are at greater risk from adverse effects from urban heat [37]. Despite being exposed to heatwave episodes for a short period of time, these heatwaves could worsen their health conditions.

The research conducted to evaluate the relationships between extreme temperatures (cold- and heatwaves), mortality [38,39,40,41,42,43], and hospitalisations [33,34,44,45,46], associating them with age groups [33,47,48] and specific diseases [45,49,50], has mainly focused on developed countries, especially those with four seasons. According to a systematic review and meta-analysis by Faurie and colleagues, a 1 °C increase in temperature raised heat-related illness hospitalisation rates by 18% [51]. Some recent studies also have found positive relationships between heatwaves and hospital admissions [33,52,53,54,55,56]. However, little research has been conducted on the association between heatwaves and hospitalisations in one-season regions. In addition, to date, there is no standard consensus or universal operational definition of a heatwave [7,42,57,58]. The impacts of heatwaves on health have been examined using a variety of heatwave definitions [38,42,47,59,60,61,62]. As a result, a better understanding of the impacts of heatwaves on the hospitalisations of the elderly in Selangor, Malaysia is beneficial for the following reasons: first, scant research has been conducted [63,64]; second, being a one-season climate, or tropical climate, implies that it experiences high temperatures throughout the year; and third, the state is populated with the highest proportion of elderly residents in Malaysia.

In order to fill the existing research gap, this study examined the impact of heatwaves on the elderly between 2010 and 2020 by age group. In addition, we examined cause-specific hospitalisations across sub-groups, including natural causes, circulatory system diseases, and respiratory system diseases, as well as by gender effects.

## 2. Materials and Methods

### 2.1. Profile of Study Area

Selangor is one of the most populous and urbanised states in Malaysia. The population of Selangor was estimated at 20.0% (or 6.5 million) of the 32.6 million in Malaysia in 2020, with 93.6% residing in urban areas and the rest in rural areas. Selangor’s residential property types range from detached, semi-detached, terrace, townhouse, and cluster properties to low-cost and flat properties, almost a quarter (24.1%) of which are low-cost and flat properties [65].

Like many other countries in the world, Malaysia is experiencing a growing ageing population, increasing to 3.5 million in 2020. Among the states and federal territories in Malaysia, Selangor had the most elderly aged 60 and older (16.4%) in 2020 [65].

In terms of healthcare facilities, Selangor is the state that has established the most hospitals (74) and medical clinics (2488), with only 18.9% public-owned hospitals and 9.3% public-owned medical clinics, compared to other states and federal territories in Malaysia [65]. Nevertheless, Malaysians and legal residents in Malaysia continue to benefit from the universal healthcare system, as the government provides a healthcare system that is one of the most affordable in the world.

### 2.2. Definition of Heatwaves

In order to take humidity into account, heatwaves were measured by their heat index (HI) value, also known as apparent temperature. In this study, a heatwave is defined as when the daily mean apparent temperature exceeded the 97.5th percentile (at least 37 °C or greater) for at least three consecutive days from 2010 to 2020.

### 2.3. Hospitalisation Data

The daily hospitalisation data from 1 January 2010 to 31 December 2020 were obtained from the Health Informatics Centre, Ministry of Health, Malaysia. We classified the data according to the International Classification of Diseases 10th revision (ICD 10) diagnosis codes: all causes (ICD10 A00-Z99), circulatory system diseases (ICD10 I00-I99), respiratory diseases (ICD10 J00-J99), and natural causes (ICD10 A00-R99). It is important to note that data for patients hospitalised for natural causes included patients hospitalised for circulatory system and respiratory diseases as well. For the elderly patients, we categorised the age groups into 60–69 years, 70–79 years, and 80 years and older, as well as 60 years and older.

### 2.4. Meteorological Data

Daily meteorological data from 1 January 2010 to 31 December 2020 were obtained from the Malaysian Meteorological Department, Ministry of Natural Resources, Environment & Climate Change, Malaysia. For this paper, we investigated the relationship between heatwaves and hospitalisations using mean apparent temperature (ATmean). Apparent temperature is a “feels like” temperature that considers the effects of humidity, and we opine that it captures more representative effects on the human body. This is relevant for Selangor in particular, which experiences high levels of daily mean relative humidity all year round. According to the NOAA’s Heat Index Chart [66], temperature can be “felt like” 43 °C (or 46 °C) with the combined effects of a temperature of 32 °C (or 34 °C) and a relative humidity of 75% (or 65%). The apparent temperature was computed utilising the following formula [67,68]:AT = −2.653 + (0.994*daily mean temperature) + 0.0153*DPT^2^(1)
where dew point temperature (DPT) = daily mean temperature − ((100-relative humidity)/5).

### 2.5. Model Specification

A Poisson regression model was adopted to investigate the association between temperature (ATmean) and hospitalisations, supporting the U-, V-, or J-shape relationships found between temperature and hospital admission rates in previous studies [44,52,69]. Generalized additive models (GAM) with the Poisson family and distributed lag models (DLM) were applied to estimate the effect of heatwave events on hospitalisations using time-series data and analysed using the statistical software R version 4.2.1 with “nlme” and “mgcv” in the GAM packages. First, the Selangor state-level effect of temperature–hospitalisation in Malaysia was examined using the model 1 as outlined below:N_t_ ~ Poisson (−µ_t_)(2)
Model 1: Ln (µ_t_) = β + π HW3D_t,_ + αATmean + S (ATmean_t_, 5) + S (DRF_t_, 5)+ S (DMEANRH_t_, 8) + δDOW_t,_ + ԑ (3)
where N_t_ is the daily hospitalisations count on day t for the elderly aged 60 and older, HW3D is a binary variable representing heatwave days when ATmean ≥ 37 °C for three or more consecutive days (HW3D = 1) or otherwise (HW3D = 0), ATmean is the mean apparent temperature, DRF is the daily rainfall, DMEANRH is the daily mean relative humidity, DOW is the day of the week, and ԑ is the error to capture the residual of the model. S are the smoothing splines with a specific degree of freedom, and Ln is the natural log; β is the intercept, and π, α and δ are coefficients.

Second, the risks of hospitalisation (RH) were computed for different age groups, stratified into 60–69-year-olds, 70–79-year-olds, ≥80-year-olds and ≥60-year-olds. Assuming a linear relationship between risk of hospitalisation and heat effects, the heat effects were modelled using interacting heatwave days (HW3D) with ATmean lag values of 0–1 day, 0–3 days, and 0–5 days, as some studies indicated that heat effects could last for at least 3 days [70]. A random effects linear meta-regression analysis was conducted to evaluate the relationship of RH with an increase of one degree Celsius during heatwave days. To investigate if gender and diseases affect hospitalisations, the variables of female (F) and male (M) and cause-specific hospitalisations, namely all causes (All_Causes), natural causes (NCause), circulatory system diseases (Cir_Sys), and respiratory system diseases (Respi) were incorporated into the model 2.
Model 2: Ln (RH) = a+ b_1_ATmean_0–1_*HW3D + b_2_ATmean_0–3_*HW3D + b_3_ATmean_0–5_*HW3D + b_4_F + b_5_M+ b_6_ LnAll_Cause*HW3D + b_7_LnNCause*HW3D + b_8_LnCir_Sys*HW3D + b_9_LnRespi*HW3D + ԑ_2_
(4)
where Ln is the natural log, a is the intercept, b_1_, …, b_9_ are coefficients, and ԑ_2_ is the error.

## 3. Results

Selangor is strategically located close to the equator region, experiencing tropical weather year-round. The monthly average weather for Selangor state from 2010 to 2020 is shown in Figure 1 and Figure 2. Figure 1 demonstrates that the daily mean temperature ranged between 27.3 °C and 28.9 °C; however, the “feels like” temperatures were higher, as recorded by the ATmean that ranged from 33.2 °C to 35.2 °C, contributed to by the high daily mean relative humidity, which varied between 75.5% and 83.2%. More importantly, on extreme hot days, “feels like” temperatures reached 44.3 °C to 47.8 °C due to extremely high daily mean relative humidity, which consistently exceeded 90%, as shown in Figure 2. Overall, the average monthly daily mean apparent temperature is on the rise on a monthly basis.

High temperatures have been associated with hospitalisation risks for elderly people who suffer from chronic diseases. Table 1 provides descriptive statistics about the monthly average hospitalisations of the elderly aged 60 and older by cause-specific disease. During the study period, the monthly average hospitalisations for all-cause diseases from the months of January to October were recorded to range between 4500 and 5600 patients, while low hospitalisation rates were recorded for November and December, with 1600 to 3000 patients. Collectively, almost 40% of hospitalised patients were diagnosed with diseases of the circulatory and respiratory systems.

For daily hospitalisations in Selangor during heatwaves and non-heatwave periods from 2010 to 2020, we estimated daily hospitalisations by cause-specific hospitalisations and age groups using a total sample of 4018 days (heatwaves, n = 35; non-heatwaves, n = 3983), as shown in Table 2. The means of daily ATmean for heatwaves and non-heatwave days were 38.1 °C and 36.0 °C, respectively, while the maximum values of ATmean were recorded as 39.0 °C and 38.4 °C, respectively. The descriptive statistics demonstrate that the means and maxima of hospitalisations during heatwaves (and non-heatwaves) for all causes ranged from 135 to 312 (154 to 473), for natural causes ranged from 124 to 294 (140 to 423), for circulatory system diseases ranged from 34 to 106 (37 to 152), and for respiratory system diseases ranged from 18 to 55 (22 to 94), respectively. For hospitalisations by age group, the mean and maximum values for those aged 60 to 69 ranged from 81 to 210 (93 to 361), for those aged 70 to 79 ranged from 66 to 615 (56 to 218), for those aged 80 and older ranged from 19 to 66 (20 to 91), and for those aged 60 and older ranged from 166 to 842 (170 to 572), respectively. Interestingly, the admissions to the hospital during heatwave days were relatively fewer than those during non-heatwave days, contrasting the expected pattern. The reason for this is that the effect of heatwave temperatures on hospitalisations of the elderly is not immediate, but could instead show lag effects on the hospitalisations. The statistics also indicate a decrease in hospitalisations by elderly age group from those aged 60 to 69 to those aged 80 and older.

During the period of this study, from 2010 to 2020, there were 35 heatwave days identified from a total of 4018 days. Table 3 indicates that a total of 2925 elderly people aged 60 and older were admitted to hospitals during the heatwave days, with 1592 (or 54.4%) of them aged 60 to 69, 985 (or 33.7%) aged 70 to 79, and 348 (or 11.9%) aged 80 and older. Analysis by cause-specific hospitalisation indicated that more than one-third of the elderly were admitted to hospitals due to circulatory system diseases (22.8%) and respiratory system diseases (13.5%).

Figure 3 shows that during heatwave days, hospital admissions were clustered around the mean apparent temperature (ATmean) ranging from 37 °C to 38 °C for all categories of diseases, namely all causes, natural diseases, circulatory system diseases, and respiratory system diseases. This can be explained by the majority (85.7%) of heatwave temperatures in Selangor state fluctuating between 37 °C and 38 °C.

After the descriptive statistics analysis, we applied a statistical modelling analysis using model 1 specified in the Model Specification section. The result in Table 4 shows that ATmean and heatwave days (or HW3D) were associated positively with admission to the hospital for the elderly aged 60 and older. Heatwave days were positively correlated with risk of admission to the hospital by 5.4% for those aged 60 and older, although this was statistically insignificant. However, an increase in ATmean by 1 °C significantly increases the risk of admission to the hospital by 12.9%.

Heat exposure was found to profoundly impact human health, as proposed by the lag models for 0 to 3 days [71]. We utilised model 2 to estimate the lag effects (for 0 to 1, 0 to 3, and 0 to 5 days) of heatwave temperatures on the elderly groups’ hospitalisation, as shown in Table 5. The results suggest that the impact of heatwave temperatures on hospitalisations for the elderly group was not immediate (refer to ATmean lag 0 to 1 day), but delayed impacts were confirmed (refer to ATmean lag 0 to 3 days) for those aged 60 to 69 and aged 80 and older. An increase of 1 °C during these heatwave days was associated with increases of 31.6% and 53.9% in risk of hospitalisation for those aged 60 to 69 and those aged 80 and older, respectively. The impact of heatwave temperatures significantly declined after 5 days (refer to ATmean lag 0–5 days), as a statistically significant negative association was identified across all age categories despite a statistically non-significant result for those aged 70 to 79.

The effects of heat on elderly hospitalisations for all-cause diseases were marginal to non-existent (−0.2% to 0.0%) during heatwave days across all age groups (Table 5). Therefore, we examined further if heatwaves cause different impacts on elderly hospitalisations due to specific causes of illness. As illustrated in Table 5, heat effects were significantly associated with higher hospitalisation rates for patients with natural causes across all age groups (aged 60 to 69: 51.4%, aged 80 and older: 46.6%, and aged 60 and older: 39.7%) during heatwave days, albeit with an insignificant increase of 20.6% among those aged 70 to 79. In addition, during heatwave days, hospital admissions for respiratory system diseases tended to increase by 48.0% (significant), 14.0% (not significant), 11.7% (not significant), and 46.3% (significant) among people aged 60 to 69, 70 to 79, 80 and older, and 60 and older, respectively. In contrast, hospital admissions due to circulatory system disease were found to be negative and significant across all age groups during the heatwave days, except for those aged 70 to 79. To investigate if gender affects hospital admissions, our results in Table 5 suggest that, compared to females (0.5% to 0.6%), males (0.4% to 0.6%) are associated with a marginally lower risk of hospitalisation rate across all age groups.

## 4. Discussion

The objectives of this paper were to investigate the impact of heatwaves on hospitalisations of the elderly from 2010 to 2020 in Selangor and to further investigate if heatwaves exert different impacts on the elderly of different age groups by number of heatwave days, specific cause, and gender.

*Age groups–number of heatwave days*: Past studies have suggested a positive correlation between high temperatures and hospital admissions [60,72]. Our findings demonstrate that during heatwave days (0 to 1 day), measured by mean apparent temperature, ATmean ≥ 37 °C for three consecutive days or more was positively associated with hospital admissions [44,53,73,74]; however, our results are not significant. The findings of past studies suggest that the impact of heatwaves on hospital admissions lasts for 3 days on average. For instance, Loughnan and colleagues examined the effects of heat on acute myocardial infarction (AMI) admissions to hospitals in Melbourne, Australia [70]. The findings suggest that when the daily temperatures exceeded the threshold temperature of 30 °C, this led to a 10% increase in AMI admissions, while an increase of 37.7% in hospital admissions occurred if temperatures exceeded 27 °C for a 3-day average. A similar finding from Tehran, Iran was that high temperatures exerted an immediate impact on current-day admissions to hospitals due to acute myocardial infarction, and this effect lasted for three days [75]. Similarly to the above studies, we observed that the impact of heatwaves on risk of hospitalisation lasted for 3 days on average, with declining effects after 5 days on average, with a negative association evidenced by a mean apparent temperature lag of 0 to 5 days.

*Age groups–specific causes*: When we correlated cause-specific hospitalisations with the heatwave variable in model 2, there was no significant result in terms of increasing hospital admissions for *all causes* across all age groups (in Table 5). Still, the impact on hospital admissions for the elderly did vary by disease. Our research results indicate that hospital admissions due to *natural causes* significantly increased across all elderly age groups while having no significant impacts for those aged 70 to 79. We also observed that during heatwave days, heat exposure significantly increased admission to hospitals due to *respiratory system* issues for those aged 60 to 69 by 48.0% and those aged 60 and older, while no statistically significant impacts were found for those aged 70 to 79 and those aged 80 and older despite positive associations. This finding is supported by Isaksen and colleagues, ref. [44] who suggested that heat exposure significantly increased hospital admissions due to heat exposure and respiratory-system-related disease in King County, Washington, USA. Similarly, in other studies, a positive relationship was identified between heatwaves and hospitalisations due to respiratory system conditions in 12 European cities [52], 114 cities in the United States [33], 2 provinces in Vietnam [55], and 1 metropolitan city in Finland [56].

Past studies showed mixed findings for impact of heat on hospitalisations due to *circulatory system* diseases. An example is a study that examined the association between heat and cold effects on hospitalisations for cardiovascular diseases in Queensland, Australia, from 1996 to 2016. The findings indicate that there was no heat effect, as a negative relationship was identified between the relative risk of hospitalisation and temperature increasing in 1995 and between 1995 and 2016, but there was a positive relationship for the year 2016 [54]. Our results in Table 5 do not support the argument that the risk of hospitalisation for the elderly with circulatory system diseases increases during heatwave days. There are some explanations for this, such as the bodies of our patients being acclimated to hot weather, as Selangor experiences tropical hot weather all year round. The use of air conditioning, electric fans, or other heat mitigation measures during heatwaves may also contribute to this. More importantly, our findings are supported by some recent studies. For instance, there was no significant positive impact of hospitalisations due to cardiovascular diseases among the elderly in the United States despite positive impacts found for hospitalisations due to renal and respiratory diseases during heatwaves [33]. Sohail and colleagues found that hospital admissions due to arrhythmia illness during heatwave days reduced by 20.8% in Helsinki, Finland [56]. In addition, a study supported that the risk of hospital admissions for elderly people (aged 60 and older) with cardiovascular diseases declined during heatwaves in Ca Mao, Vietnam [55]. Conversely, some studies found a positive heat impact on hospitalisations due to circulatory system diseases in New York City, USA [69] and Queensland, Australia [54].

*Age groups–gender*: Table 5 demonstrates that females were marginally more vulnerable to heatwave hospitalisations compared to males across all age groups. This is consistent with Phung et al.’s research, which indicates that heatwaves caused higher all-cause hospitalisation risk among females in Vietnam [76]. Despite there being some studies that found that males are at relatively higher risk than females [53,54], a number of factors could contribute to these differences, including heatwave-related risk perceptions, physiological differences, living environments, personal behaviours, socioeconomic status, and activities. According to Conti et al.’s systematic review, heatwave-related risk perceptions differ according to demographic, social, and economic factors, and future research should focus on mixed methods approaches [57]. Some scholars found that the elderly did not perceive themselves as being at high risk for heatwaves [77,78].

Overall, our research is the first study that explores the association between heatwaves and the elderly in Malaysia despite there being few studies related to heatwaves. The evidence from this research suggests that the elderly in Selangor, particularly females, are vulnerable to high ambient temperatures, albeit this impact could vary for people with different diseases. Thus, our research is worthwhile to be used as a guide in improving public health strategies to target older adults who are at the greatest risk of hospitalisations related to heatwaves. It also provides an important step in developing early severe weather and health warning systems that could facilitate a reduction in severity of the heatwave-related health risks of elderly people. This will, in turn, reduce the burden on the hospital management system in Selangor state and Malaysia in general.

Several limitations were present in this study. First, the data used in this study cover only Selangor state, as Selangor (with 16.4% of the total elderly population) was among the states with the highest elderly proportion in Malaysia in 2020. Therefore, the findings of this study cannot be generalised to represent the whole of Malaysia, as the results could potentially be different due to the heterogeneity across states within the country. Second, due to data availability, this study did not examine individual characteristics such as the elderly patients’ socioeconomic status, including their economic status, education level, ethnicity, and whether they were staying alone or with family members.

## 5. Conclusions

Overall, our study shows that heatwave days are positively associated with hospital admissions of those aged 60 and older, albeit not significantly. More importantly, our findings indicate that an increase in mean apparent temperatures by 1 °C significantly increases the risks of admission to a hospital by 12.9%. The studied heatwaves did not exert an immediate impact on hospital admissions of the elderly; rather, they had delayed effects on hospitalisations. However, hospital admission rates of the elderly groups were found to start declining after an average of 5 days following heatwaves. While heatwaves increased hospital admissions of the elderly with natural causes and respiratory system diseases, opposite effects were found for the elderly with circulatory system diseases. The evidence in this study indicates that elderly females are marginally more vulnerable than males.

## Figures and Tables

**Figure 1 ijerph-20-05910-f001:**
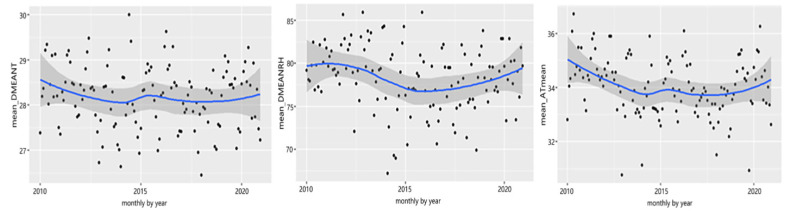
Average monthly daily mean temperature (mean_DMEANT, °C), daily mean relative humidity (mean_DMEANRH, %), and daily mean apparent temperature (mean_ATmean °C), 2010 to 2020.

**Figure 2 ijerph-20-05910-f002:**
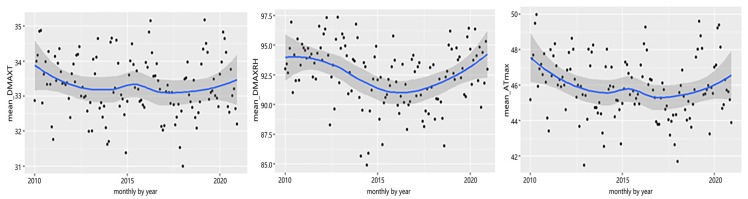
Average monthly daily max temperature (mean_DMAXT, °C), daily max relative humidity (mean_DMAXRH, %), and daily max apparent temperature (mean_ATmax, °C), 2010 to 2020.

**Figure 3 ijerph-20-05910-f003:**
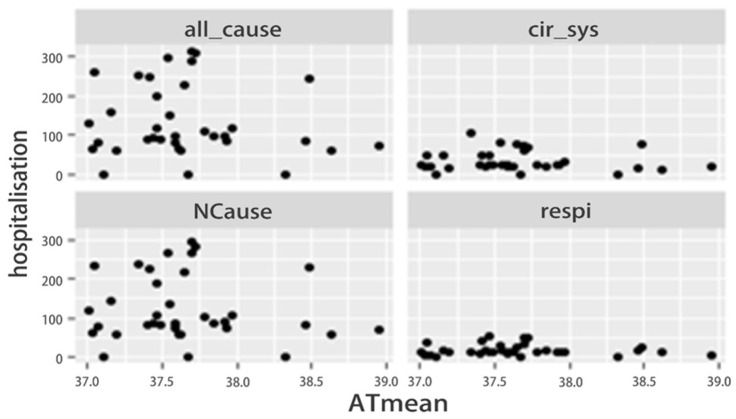
Heatwaves and Hospitalisations by Type in Selangor, 2010 to 2020.

**Table 1 ijerph-20-05910-t001:** Summary of the monthly average elderly’s (aged 60 and older) cause-specific hospitalisations in Selangor, 2010–2020.

Month	Hospitalisation							ATmean (°C)	
	All-Cause Diseases	Natural Causes	Share (%)	Circulatory System ^1^	Share (%)	Respiratory System ^2^	Share (%)	Mean	S.D.
Jan	5590	5133	91.8	1318	23.6	890	15.9	33.2	1.5
Feb	4924	4507	91.5	1164	23.6	802	16.3	33.7	1.1
Mar	5342	4874	91.2	1247	23.3	841	15.7	34.5	1.2
Apr	5058	4624	91.4	1230	24.3	742	14.7	35.0	1.4
May	5333	4874	91.4	1299	24.4	751	14.1	35.2	1.4
June	5193	4730	91.1	1257	24.2	703	13.5	34.8	3.1
July	5265	4816	91.5	1260	23.9	721	13.7	34.3	1.2
Aug	5266	4821	91.5	1285	24.4	729	13.8	34.3	1.2
Sep	4943	4525	91.5	1193	24.1	687	13.9	33.6	2.3
Oct	4533	4138	91.3	1073	23.7	630	13.9	33.5	2.9
Nov	2987	2722	91.1	723	24.2	425	14.2	33.4	1.1
Dec	1624	1429	88.0	376	23.1	243	14.9	33.0	2.7

Source: Ministry of Health, Malaysia, author’s calculation. Note: ^1^ Exclude: chronic rheumatic heart diseases (I05–I09), ^2^ Exclude: lung diseases due to external agents (J60–J70).

**Table 2 ijerph-20-05910-t002:** Statistics of Daily Hospitalisations in Selangor during Heatwaves and Non-Heatwave Periods, 2010–2020.

	Heatwaves (n = 35)					Non-Heatwave (n = 3983)				
Hospitalisation	Mean	Max	Q3	Median	Q1	Mean	Max	Q3	Median	Q1
All-cause	135	312	212	98	78	154	473	250	119	71
Natural cause	124	294	202	88	72	140	423	228	108	65
Circulatory system	34	106	49	24	20	37	152	58	27	16
Respiratory system	18	55	21	12	11	22	94	36	17	10
Age 60–69	81	210	118	62	44	93	361	149	69	44
Age 70–79	66	615	87	35	32	56	218	88	44	29
Age 80 and older	19	66	28	12	10	20	91	32	15	9
Age 60 and older	166	842	234	109	81	170	572	275	127	85
ATmean (°C)	38.1	39.0	37.8	37.6	37.4	36.0	38.4	35.0	34.1	33.1

Note: heatwaves defined as when daily mean apparent temperatures (ATmean) ≥ 37 °C for at least three consecutive days. Q1: first quartile, Q3: third quartile, SD: standard deviation.

**Table 3 ijerph-20-05910-t003:** Descriptive Statistics of Admission to Hospitals by Age Group and Cause-Specific Hospitalisations during Heatwaves, 2010 to 2020.

Year	Date of Start	Heatwaves (Days)	Age Groups				Diseases			
			60 to 69	70 to 79	80 and Older	60 and Older	All Causes	Natural Causes	Circulatory System	Respiratory System
2010	May-15	3	155	86	32	273	273	245	55	27
2010	May-23	5	272	169	52	493	493	441	116	62
2011	May-06	5	194	122	48	364	364	319	77	37
2012	April-27	4	158	140	47	345	345	266	85	35
2012	May-14	3	196	114	38	348	348	278	77	40
2012	June-02	4	147	76	23	246	246	200	53	42
2016	April-12	3	470	278	108	856	856	754	160	126
2020	May-05	8	1255	1328	305	2888	2888	1831	581	245
	Total	35	1592	985	348	2925	2925	2503	623	369

Note: heatwaves defined as if ATmean ≥ 37 °C or greater for at least 3 consecutive days or more.

**Table 4 ijerph-20-05910-t004:** The Association between Temperatures and Hospitalisations for the Elderly in Selangor, 2010–2020.

Parametric Coefficients	Estimate	Std. Error	t Value	Pr (>|t|)	
(Intercept)	0.130	0.215	1.360	0.187	
ATmean	0.129	0.002	64.319	0.000	***
Heatwave	0.054	0.313	0.173	0.863	
Day of Week					
Mondays	0.195	0.097	2.017	0.044	*
Tuesdays	0.124	0.097	1.285	0.199	
Wednesdays	0.160	0.097	1.660	0.097	#
Thursdays	0.048	0.096	0.494	0.622	
Fridays	0.028	0.097	0.385	0.088	
Saturdays	0.000	0.097	-0.001	0.999	
Sundays	0.035	0.097	0.360	0.719	
Approx. sig. of smooth terms:	edf	Ref.df	F	*p*-value	
s(ATmean)	3.536	3.816	19.673	0.000	***
s(DRF)	2.394	2.879	6.092	0.000	***
s(DMEANRH)	3.593	4.365	35.752	0.000	***

Sig. codes: *** *p* < 0.001, * *p* < 0.05, # *p* < 0.1.

**Table 5 ijerph-20-05910-t005:** The Impacts of Heatwaves on Hospitalisation Rate by Age Group, 2010–2020.

	Aged 60 to 69		Aged 70 to 79		Aged 80 and Older		Aged 60 and Older	
(Intercept)	−0.161	#	−0.800	***	−0.738	***	0.335	***
ATmean								
Lag 0–1 day	0.013		0.104		−0.164		0.032	
Lag 0–3 days	0.316	#	−0.127		0.539	*	0.213	
Lag 0–5 days	−0.530	**	−0.156		−0.417	#	−0.432	**
Gender								
Female	0.005	***	0.006	***	0.006	***	0.006	***
Male	0.005	***	0.005	***	0.004	***	0.006	***
Hospitalisations								
All causes	−0.001		0.000		0.000		−0.002	*
Natural causes	0.514	**	0.206		0.466	*	0.397	**
Circulatory system	−0.953	***	−0.430		−0.781	*	−0.666	**
Respiratory system	0.480	***	0.140		0.117		0.463	***

Sig. codes: *** *p* < 0.001, ** *p* < 0.01,* *p* < 0.05, # *p* < 0.1.

## Data Availability

The data presented in this study are available upon reasonable request from the corresponding author. In order to protect privacy and health information, the data are not available publicly.

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
