# Peer review of "The Scorching Truth: Investigating the Impact of Heatwaves on Selangor’s Elderly Hospitalisations"

_ijerph, 2023, doi:10.3390/ijerph20105910_

Round 1
Reviewer 1 Report
Thanks for the opportunity to review this paper, an observational study using meteorological and hospital admission data over 11 years in Selangor, Malaysia.
The study showed an increase in hospitalisations during the three days following a heatwave for two of the three age subgroups of elderly patients. Heatwaves were associated with an increased in respiratory-related admissions but a decrease in admissions related to the circulatory systems, albeit with these not reaching statistical significance for some subgroups.
Data were always presented with three age subgroups (60-69, 70-79, 80+) with no overall data presented.
Some comments below:
- Could overall results for all patients >=60 years have been presented as well? Given the very small number of heatwaves days over the study period (35 days out of 4000), is it possible that dividing the population into smaller age groups could have diluted an overall true effect and led to some non-significant findings or discordant findings (e.g. negative relationship for 70-79 group when positive elsewhere) in Table 5?
- In Table 4, the total number of patients doesn’t equal the number in the ‘All cause’ column. Can the authors provide an explanation in the text as to why this is so? Was there missing data?
- Similarly, there appears to be significant overlap between natural causes, circulatory system and respiratory system for reason for admission in Tables 2 and 4. If so, can the authors be explicit in their Materials and Methods section 2.1 that patients can be counted in more than one group (for those unfamiliar with ICD groupings)?
- Table 5 – analysis of causes for hospital admissions – this appears to have only been assessed for one scenario with regards to lagged effect. Is it possible to assess the lagged effect (if any) of heatwaves on admissions for specific causes? For instance, some patients with heart failure or renal impairment may have some clinical reserve, and initially avoid hospitalisation after a heatwave but after a few days their reserve may be overcome, and admission is necessitated. Could such an analysis for lagged effect of cause specific admissions (even if on all age groups) be performed? Potentially as supplementary materials if not significant.
- Would be good to have more context about Selangor for international readers not familiar with this region or Malaysia more broadly – what is the socio-economic status of Selangor residents? What type of housing do most residents have? What proportion of those >60 years live at home, versus in a nursing home or care facility? What is the prevalence of air conditioning or fans within homes, compared to nursing homes/care facilities? Is there universal healthcare coverage for citizens (which could impact on the likelihood of presenting for healthcare)? Region-level data may not be available for all the above, but where possible this context would be helpful to understand the background vulnerabilities and protective factors for residents of Selangor. Some of these issues were acknowledged in the Limitations, and given the observational nature of the study it is understandable that these data were not available for individuals admitted to hospital, but giving information about the typical living conditions in Selangor will provide helpful context for interpreting your results.
- Discussion section could talk more about risk perception among the elderly. Conti et al 2022 systematic review on Knowledge Gaps and Research Priorities on the Health Effects of Heatwaves emphasised that the elderly were considered to be a group with a particularly low-risk perception, despite being a high-risk population.
- Females appear to be at higher risk in Selangor, in contrast to the published literature from elsewhere. Do the authors have any thoughts about why that might be? Any activities/exposures that are more associated with women rather than men in Selangor?
Reviewer 2 Report
This study, while limited to a specific region of Malaysia having a fairly limited year-round temperature band, brings forward some counterintuitive conclusions about hospitalization risk under heatwave days. The statistical analysis appears sound, though it is densely presented with too many acronyms and hard to follow in current narrative form. Generally, the literature backs up the authors’ presentation that the association between extreme heat and hospitalizations has not been as well tracked as mortality, but no reason is not provided (mostly because administrative data is less available on hospitalizations among the elderly). Some more recent references and acknowledgement that there may be different opinion on defining heatwaves, definitions do exist. Make the case for what definition this manuscript will use, as currently that is not stated. Also, the tables average heat across 2010-2020, so we don’t actually know if heat risk is growing stronger with time.
Overall, the manuscript itself needs to provide smoother narrative transitions, and not just descriptive stats. More attention needs to go toward the argument itself for why parsing out hospitalizations by circulatory vs respiratory causes is important. How and where can this study be generalized, given rising temps, should also be stated.
Abstract:
L14: remove “the” before hospital systems
L 42-44: references to heat and mortality are dated. Please provide more recent citations, e.g., Gasparrini.
L48: These articles don’t explicitly say that heatwave is not defined. Liss [20] for example, doesn’t provide a clear definition. Definitions of heatwaves and heatwave indices do exist in the literature. See for example:
Retrospection of heatwave and heat index | SpringerLink
Heat Wave and Mortality: A Multicountry, Multicommunity Study | Environmental Health Perspectives | Vol. 125, No. 8 (nih.gov)
A comparison of heat wave climatologies and trends in China based on multiple definitions | SpringerLink
Ln 55: Your RQ currently is stated as: the article investigates “the impact of heatwaves on the elderly by age group.” The RQ would benefit from a more precise association than impact—hospitalizations, emergency calls, etc. and by “decadal” age group, or “age group by decade”.
Ln 87: change “was” to “were”
Ln 94: place DOW in proper sequence when describing variable acronyms
Model 2 seems overfitted.
Table 2: Not needed. If you keep, please remove % share of all-cause diseases column.
Ln 130-131: Grammar: We estimated daily hospitalizations by….
Ln 133-134: here you provide mean aTemp of heatwave and non-heatwave days. Please provide a benchmark earlier in paper for your definition of heatwave and what mean temp you will be using as a reference for your results.
Ln 136-138: I don’t see the corresponding entries for these number ranges in Table 3.
Ln 138-140: Similarly, I don’t see the corresponding entries for these number ranges in Table 4.
Fig 1: please make clearer.
Ln 169: does this paragraph description correspond to a table? I don’t see your results (“the result” described in Ln 170 doesn’t have a tabular entry).
Ln 188-194: vague language. This association is not made clear as to how it relates to heat waves.
Table 5: column 1 variables need clearer explanations; currently too cryptic to make any sense of them.
Ln 242: confusing explanation of hospitalization and circulatory systems.
Ln 292-295: your study did not control for socioeconomic and demographic variables. This is true whether or not you examined the findings by states, as those variables are not dependent on state-level comparisons, so you could examine SD vars but chose not to.
Limitations: suggest that you place these at end of the Discussion section, not following Conclusions.
You should speak of habituation of Selangor’s residents to high levels of “feels like” heat in your discussion, which is why hospitalizations show a negative relationship to heatwave days. Did you also look at use of air conditioning, government shelters, or other mitigative health measures to protect them from heat?
Use of English language is overall sound. However, there are specific passages where imprecision gets in the way of understanding your points. Please see above comments for specific examples of where to improve.
Reviewer 3 Report
Thank you for the opportunity to review this manuscript. The authors have addressed an important issue related to rising ambient temperatures/heatwaves and associated hospitalizations for older adults. I appreciate the use of ICD10 codes for addressing this important issue, they need to be used more frequently in research to address environmental health issues. I have a few recommendations to strengthen the overall merit of this paper:
1) In the abstract is it stated that hospital admissions are "insufficiently studied (line 12)", however in the introduction you state that it has “been well researched (line 44)." Please make this uniform.
2) In the discussion section, lines 270- 271. I recommend adding: to improving public health strategies to target older adults at greatest risk of hospitalizations related to heatwaves.
3) Move the limitations to the end of the discussion, you don't want to end with a negative, when the study supports valuable information.
4) line 290, make finding plural to be: findings.
Well written. One comment above related to the word finding.
